# Antibiotherapy in Children with Cystic Fibrosis—An Extensive Review

**DOI:** 10.3390/children9081258

**Published:** 2022-08-20

**Authors:** Ioana Mihaiela Ciuca, Mihaela Dediu, Diana Popin, Liviu Laurentiu Pop, Liviu Athos Tamas, Ciprian Nicolae Pilut, Bogdan Almajan Guta, Zoran Laurentiu Popa

**Affiliations:** 1Pediatric Department, University of Medicine and Pharmacy “Victor Babes” Timisoara, 2 Eftimie Murgu Square, 300041 Timisoara, Romania; 2Pediatric Pulmonology Unit, Clinical County Hospital Timisoara, Evlia Celebi 1-3, 300226 Timisoara, Romania; 3Biochemistry Department, University of Medicine and Pharmacy “Victor Babes” Timisoara, 2 Eftimie Murgu Square, 300041 Timisoara, Romania; 4Microbiology Department, University of Medicine and Pharmacy “Victor Babes” Timisoara, 2 Eftimie Murgu Square, 300041 Timisoara, Romania; 5Kinesiotherapy and Special Motricity Department, West University of Timisoara, 4 Vasile Parvan bld., 300223 Timisoara, Romania; 6Department of Obstetrics and Gynecology, University of Medicine and Pharmacy “Victor Babes” Timisoara, 2 Eftimie Murgu Square, 300041 Timisoara, Romania

**Keywords:** cystic fibrosis, antibiotics, *Pseudomonas aeruginosa*, *Burkholderia cepacian* complex, personalized antibiotherapy

## Abstract

In cystic fibrosis (CF), the respiratory disease is the main factor that influences the outcome and the prognosis of patients, bacterial infections being responsible for severe exacerbations. The etiology is often multi-microbial and with resistant strains. The aim of this paper is to present current existing antibiotherapy solutions for CF-associated infections in order to offer a reliable support for individual, targeted, and specific treatment. The inclusion criteria were studies about antibiotherapy in CF pediatric patients. Studies involving adult patients or those with only in vitro results were excluded. The information sources were all articles published until December 2021, in PubMed and ScienceDirect. A total of 74 studies were included, with a total number of 26,979 patients aged between 0–18 years. We approached each pathogen individual, with their specific treatment, comparing treatment solutions proposed by different studies. Preservation of lung function is the main goal of therapy in CF, because once parenchyma is lost, it cannot be recovered. Early personalized intervention and prevention of infection with reputable germs is of paramount importance, even if is an asymmetrical challenge. This research received no external funding.

## 1. Introduction

Cystic fibrosis (CF) is the most common potentially fatal monogenic disease in the Caucasian population, with autosomal recessive transmission and a remarkable clinical polymorphism [1]. Typical CF manifestations include obstructive lung disease and chronic diarrhea, with several complications such as liver and bone disease, CF-related diabetes, which can occur in time [2].

The severity of CF lung disease predicts the life expectancy in patients, and its evolution is marked by multiple exacerbations. Therefore, the maintenance of a stable, functional, and germs-free lung is the mainstay in CF therapy. Even with significant improvement in CF patient’s life expectancy [3], the respiratory failure remains the final stage of CF patients’ life. It became mandatory to slow lung disease progression as a primary CF therapy target [4].

From pathological point of view, CF is associated with the failure of muco-ciliary clearance with subsequent mucus plugging and frequent infections with opportunistic microorganisms. Commonly encountered pathogens, “traditionally” CF associated, are *Pseudomonas aeruginosa* and *Staphylococcus aureus*, and relatively “infrequent” but very redoubtable germs are represented by the complex *Burkholderia*, *Stenotrophomonas maltophilia*, *Achromobacter xylosoxidans,* or *non-TB mycobacteria* [5]. The consequence of lung colonization and infection is a neutrophil-driven inflammation with frequent exacerbations that, in time, leads to a significant loss of lung function, which will never fully recover to its previous condition [6]. Respiratory pathology is the determining element of CF patients’ evolution, lung infections being the main cause of morbidity and mortality [7].

Recently, CFTR modulator therapy showed promising results in slowing the disease or even stopping the damaging effects [8]. However, even so, until the discovery of a specific targeted therapy for every pathogenic mutation, the lung must be treated and its function carefully preserved. Therefore, besides clearance techniques and mucolytic therapy, the antibiotherapy plays a crucial role in CF lung disease management and life expectancy improvement.

Over the years, several pathogen-specific antibiotics were used in CF lung disease therapy, however this therapeutic experience was included in very few guidelines [7,8,9]. On the one hand, there were a reduced number of studies with a significant number of samples and on the other hand, the distinct response to treatment of each patient imposed an individual-specific, personalized antibiotherapy.

The main goals of antibiotic therapy in CF are the prevention, eradication, and control of CF-associated respiratory infections. General principles of antibiotic therapy are to introduce aggressive treatment from the beginning, in higher doses than in other conditions, with prolonged cures of 2–4 weeks, and association of the nebulized antibiotics [7,10,11]. Guided treatment depends on the severity of the symptoms, patient status, and finally but most important on the isolated germ.

The aim of this paper is to present current existing antibiotherapy solutions for CF-associated infections in order to offer a reliable support for individual, targeted, and specific treatment.

## 2. Materials and Methods

This review was performed in accordance to the PRISMA (Preferred Reporting Items for Systematic Reviews and Meta-Analyses) guidelines. For the current review, the PubMed and ScienceDirect databases were searched, focusing on articles written in English.

### 2.1. Search Strategy

The search strategy was designed to address the following:Population: patients with CFIntervention: antibiotic treatment of respiratory exacerbationsComparisons: literature researchOutcome: accuracyStudy design: any

We conducted the literature search in PubMed and ScienceDirect, until December 2021, covering a 30-year period that marked the evolution of CF antibiotic treatment. The following keywords were used: cystic fibrosis/infections/antibiotics, with focus on the following search string ((cystic fibrosis) AND (infections)) AND (antibiotics). The search strategy was designed and carried out with input from all investigators.

### 2.2. Inclusion Criteria

Original articles, special articles, systematic reviews were taken into consideration for the systematic review.Studies involving pediatric CF patients with recurrent infections with various types of germs.Only studies involving the cystic fibrosis/antibiotics /infections were included.Studies written only in English were taken into consideration.

### 2.3. Exclusion Criteria

Studies that discussed the same topic, which proved to have the same hypothesis and results were not included in the paper, being included only a limited number of studies that presented superior results to the others.Studies involving adult CF population were not taken into consideration for the research of this review.Studies that presented only in vitro results, without clinical outcome, were also not included.

## 3. Results

A total number of 15.979 studies were identified from two scientific databases: PubMed and ScienceDirect, but 13.467 of records were removed because of the inability to download more than 1000 reference records from ScienceDirect, so they were subtracted. For the duplicate removal, we used the Endnote reference manager, resulting a total number of 214 duplicates, which were removed from the total number of the records, resulting 2405 records to be screened. After screening the titles and the abstracts of the identified records, only 622 records were eligible and, finally, only 206 of these records had full text. In the final step, we searched carefully which of the studies we found met our inclusion criteria, and after this selection only 74 records qualified to be used in this review: 35 original articles, 33 review articles, 2 clinical trials, 1 meta-analysis, 1 guideline, and 2 case reports with a total number of 26,979 patients aged between 0 and 18 years. The flowchart below shows the steps that have been followed and the methodology used in deciding upon the eligible articles (Figure 1).

### 3.1. Antibiotic Treatment in CF Patients

#### 3.1.1. *Pseudomonas aeruginosa* Infection

Though the spectrum of cystic fibrosis “specific” pathogens is significant, the most common is *Pseudomonas aeruginosa*; its acquisition and persistence is associated with lung deterioration and progression of the lung disease, frequent exacerbations, and increased mortality [12]. Antibiotic treatment is the mainstay therapy for CF lung disease with proven efficacy, improving health and prognosis of people with CF [13].

Several therapeutic regimens were proposed, having a good and solid efficacy, but still, an individualized approach is necessary in order to address specific characteristics of each CF patient that can influence the response to general therapeutic regimens. The antibiotic choice differs, depending on the infection moment, the lung colonization, which will occur in time and the mucoid transformation of the *Pseudomonas aeruginosa* [14]. As *Pseudomonas aeruginosa* infection is the major cause of morbidity and mortality in patients with CF [15], the patient’s evolution largely depends on the early identification and prophylactic treatment of chronic infection [16] and risk factors eviction [17].

The *Pseudomonas aeruginosa* infection occurs in the majority of patients [18], and has different characteristics and consequences, according to the moment of infection, antibiotic strategy, and response to treatment [19]. If in the first positive culture, the germ is mobile, with low density and the nonmucoid strains have not developed biofilm, than at this stage sterilization is still possible [14]. The initial infection with the microorganism, usually in planktonic form, will be followed by chronic infection (mucoid *Pseudomonas aeruginosa*). In the planktonic form, antibiotics can eradicate the microorganism [20]; however, persistent infection is associated with biofilm growth and adaptive evolution mediated by genetic variation. The factors linked with persistent infection and problematic eradication are the development of mucoid strains with a high rate of adaptive mutation and antibiotic resistance [21]. There is clear evidence of the possibility of eradicating the first pseudomonal infection, but no concluding data on the superiority of one therapeutic regimen over another [22].

Chronic *Pseudomonas aeruginosa* infection is accompanied by an increased resistance to antibiotics, greater frequency of exacerbations, loss of lung function, and shortened lifespan [23]. *Pseudomonas aeruginosa* infection is defined as: intermittent, when the cultures are positive in less than 50% of the annual cultures and chronic if more than 50% of the annual cultures are positives [19].

*Pseudomonas aeruginosa* infection is treated with specific antibiotics [24], in combination of two or more, the mode of administration and treatment duration depend on the clinical condition, the severity and duration of the infection, the response to therapy, the aggressiveness of the strain, and individual response to treatment [25]. The analyzed data did not show a synergistic effect of certain antibiotic combination, concluding that the non-bactericidal effects of antibiotic therapy might have an important effect of the combined therapeutic effect [22].

In a recent Cochrane review, Langton and Smyth showed that nebulized antibiotics, alone or in combination with oral antibiotics have a better effect than no treatment for early infection with *Pseudomonas aeruginosa*, comparing nebulized tobramycin, versus oral ciprofloxacin with nebulized colomycin and oral ciprofloxacin with nebulized tobramycin [18]. There was insufficient evidence to show a difference in rates for eradication of early *Pseudomonas aeruginosa* between the different antibiotics regimens, in order to state which antibiotic strategy should be used for eradication of primary *Pseudomonas aeruginosa* infection in CF patients [18].

For the first pseudomonal infection, the therapeutic options are represented by administration of oral ciprofloxacin combined with inhaled tobramycin or inhaled colomycin [7]. The Early Pseudomonas Infection Control (EPIC) trial found that a comparable percentage of children remained free of *Pseudomonas aeruginosa* in groups that received inhaled tobramycin with and without oral ciprofloxacin [7]. In moderate/severe infections, intravenous therapy in hospital would have better outcome. Combination of beta lactams with aminoglycoside would be of first choice, while imipenem or colomycin with aminoglycosides of secondary choice, as aminoglycosides are a drug class with a satisfactory sputum concentration [17,26].

Bacteriological evaluation followed by subsequent therapy strategy is necessary: antibiotics administration if the culture is positive and monitoring the respiratory cultures at one month [27]. If the culture is negative, inhaled tobramycin or colomycin would be the treatment of choice. If the culture is positive and the infection is not eradicated, an individualized treatment should be instituted, until eradication is obtained. Failure to eradicate the primary infection with *Pseudomonas aeruginosa* leads to chronicity. The USA guidelines endorse inhaled antibiotics therapy for patients with chronic *Pseudomonas aeruginosa*, continued forever and the European recommendations are similar, either single drug therapy or alternating therapy of different antibiotics, especially in those patients with moderate/severe lung dysfunction [27]. As an alternative, inhaled colomycin can be used in continuous administration, and nebulized aztreonam has shown good efficacy, comparable to tobramycin administration [28].

Individualized antibiotherapy has several specific aspects especially in children, where some antibiotics are age limited, such as inhaled tobramycin—restricted to children older than 6 year [29].

Different studies showed that other inhaled drugs such as liposomal amikacin, levofloxacin, fosfomycin/tobramycin, or ciprofloxacin have a significant effect on chronic infection treatment. In addition, some studies suggested that aztreonam lysine for inhalation (AZLI) has proven to be superior over nebulized tobramycin in a chronic infection with *Pseudomonas aeruginosa* [30] having a high concentration in the sputum, reducing the bacterial density [31].

Chronic infection treatment currently uses inhaled therapeutic regimens, as well as chronic oral azithromycin therapy, effective in reducing the number of exacerbations and maintaining lung function, even improving basal ventrilo-metric parameters [32].

The antibiotic pharmacokinetics differ individually, therefore the choice of antibiotic dosages used in therapy has to be made according to guidelines and centers experience [33].

For *Pseudomonas aeruginosa*, a combination of two or more antibiotics is recommended and, although evidence is lacking, two weeks of intravenous treatment is routine [9] with the possibility of prolongation for patients requiring individualized approach [34].

The choice of antipseudomonal therapy can be done considering the empiric advised susceptibility or according to in vitro susceptibility, determined by respiratory cultures, which is a generally applied strategy for severe respiratory infections [11]. In cystic fibrosis it has been showed that the clinical outcome of infected CF patients and in vitro susceptibility data are poorly correlated [22]. Therefore, the success rate of a given antibiotic treatment course using microbiologically driven antibiotic treatment data is frequently failing [35], while antibiotics regimens associated with clinically obvious effects were noticed to have a better effect [36].

In some centers, regular antibiotic treatment was practiced, intravenously, every 3 months, which seemed to have good efficacy, being somewhat superimposable with the treatment of exacerbations [37]. In case of an exacerbation, it is recommended to combine two parenterally administered antipseudomonal antibiotics, β lactamases or carbapenems, for a period of 14 days, depending on the clinical condition, associated with 3 months tobramycin or colomycin [38,39]. Regular intravenous antibiotic treatment at 3 months might be recommended in cases of moderate/severe respiratory dysfunction and chronic respiratory failure by certain centers from northern Europe. This antibiotic therapy regimen is regularly practiced at 3 months in Danish and Norwegian centers, resulting in increased life expectancy of CF patients and the lowest percentage of chronic *Pseudomonas aeruginosa* infection [37].

A recent study conducted by Frost et al. showed the benefits of inhalation of aztreonam lysine (AZLI) with intravenous therapy using 14 days of inhaled aztreonam lysine plus intravenous (*i.v.*) colistimethate (or dual *i.v.* antibiotics), which included meropenem, ceftazidime, or piperacillin/tazobactam. According to Frost, after 14 days of treatment, AZLI+IV was associated with an improvement in lung function (estimated to an extra 4.6% predicted FEV1) comparing to the association with IV+IV [30].

A recent important study, TORPEDO-CF failed to demonstrate the superiority of intravenous treatment over oral therapy in achieving the *Pseudomonas aeruginosa* eradication at 3 months and being infection free up to 15 months, but the number of admissions in intravenous-treated patients was significantly lower compared to orally treated patients [18].

Lately several antibiotics such as ceftazidime/avibactam and ceftolozane/tazobactam showed good in vitro liability [40], and valuable options for CF exacerbations with *Pseudomonas aeruginosa* for adults [41] were reported in several case reports [42,43], but there are not reliable studies regarding these antibiotics’ regimens in pediatric CF patients.

All the antibiotics regimens are presented briefly in Table 1.

#### 3.1.2. *Staphylococcus aureus* Infection

It is the most common germ isolated in the sputum of children with CF in the first decade of life and has been supposed to be a precursor of later infection by *Pseudomonas aeruginosa* [11]. The staphylococcal infection is associated with increased lower airway inflammation.

*Methicillin-sensitive Staphylococcus aureus* (*MSSA*) is a challenging germ, with increasing prevalence, the source of infection being nosocomial but also community, and the infection occurs especially in patients with poor lung function.

The presence of a positive culture with *Staphylococcus aureus* divides cystic fibrosis patients in four categories: those with a first infection who can be symptomatic or without symptoms, and those patients with chronic infection but clinically stable or with an acute respiratory exacerbation [45].

Although the germ is known to be the main cause of infectious lung exacerbations and lung function decline [46], the studies conducted so far could not establish if the prevention of colonization with this germ can lead to a better clinical outcome.

In the United Kingdom, the Cystic Fibrosis Trust Antibiotic Working Group recommends in 2017 the initiation of anti-staphylococcus prophylaxis with antibiotics with a narrow spectrum of action, started from the neonatal period up to the age of 3, or in some cases up to 6. The decision is based on a several studies carried out some time ago, by Loening-Baucke et al. (1979) and Weaver et al. (1994), which highlighted the fact that the patients that followed a prophylactic treatment with cefalexin for 2 years had a lower rate of respiratory problems and needed fewer days of hospitalization and antibiotic treatment. In case of patients who did not benefit of prophylactic treatment, a higher number of *Staphylococcus aureus* infections, more hospitalizations days, and a higher need of antibiotic treatment for every episode were registered.

Separate studies, from Germany and United States, could not demonstrated the efficacy of prophylaxis with oral cephalosporines: patients did have fewer number of sputum culture with *Staphylococcus aureus* but a higher number of culture positive for *Pseudomonas aeruginosa* [47,48].

The discussion regarding *Staphylococcus aureus* infection is not only about if to prescribe or not a prophylactic treatment. At the moment there is no official protocol about the therapy in a first infection. The only certitude that exists is the fact that antibiotics should be administrated in order to treat an acute *Staphylococcus aureus* infection [49].

Concerning a treatment schedule of acute *Staphylococcus aureus* infection in symptomatic patients, in most of the cases it depends on the attitude and experience of the doctor and on germ sensibility to different antibiotics. The first attempts to treat this infection consisted in the association of two antibiotics, in general, flucloxacillin and rifampicin or fusidic acid. The duration of this treatment is between 2 and 4 weeks, and for patients who do not respond, a new cure can be administrated [50].

This treatment schedule is also proposed by UK Cystic Fibrosis Antibiotic Working Group, with the mention that in case of no response after the first cure of oral antibiotics therapy, intra venous antibiotics can be administrated, according to the antibiogram [36].

All the antibiotics regimens are presented briefly in Table 2.

#### 3.1.3. Methicillin-Resistant *Staphylococcus aureus* (MRSA)

If the patient is diagnosed for the first time with *MRSA* infection or the infection appears in a patient who was initially declared *MRSA* free, the therapy purpose is to eradicate the infection. Contrary to *Pseudomonas aeruginosa* infection, in *MRSA* infection there is no accurate definition to attest the status of chronic infection. Studies about *MRSA* infection treatment, had defined as chronic infection: “the presence of at least three positive cultures with *MRSA* in the last 6–12 months” [51,52]. The therapy includes standard topical treatment and oral antibiotics (rifampicin and fusidic acid) or vancomycin in aerosol-therapy, or all of these combined. Intravenous treatment with vancomycin is the standard therapy for acute *MRSA* infection, but its respiratory efficacy is reduced due to its low penetration into the lung secretions and also due to nephrotoxicity, which imposes limits on the dose of vancomycin administered. Thus, through inhalation therapy, higher concentrations of antibiotics in the lungs might be achieved, reducing the risk of developing systemic side effects. Over time, vancomycin has been used by inhalation as an off-label drug. A study published by Dezube et al. reports that the use of inhaled vancomycin reduced the number of *MRSA* colonies, but do not eradicate them [53]. The study published by Waterer et al. shows the development of a type of dry vancomycin powder for inhalation (AeroVanc), which has a high sputum concentration, but being a first phase study it requires additional steps to build the utility of AeroVanc powder [52].

Fosfomycin/tobramycin was used in past as aerosol-therapy to treat infection with germs as: anaerobes, Gram negative or Gram positive. At present, vancomycin is also used in aerosol-therapy in order to eradicate *MRSA* infection, with the mention of using a bronchodilator before. Studies had demonstrated the safety and well tolerance of this therapy [54].

A multicentric study performed in Italy by Dolce and colleagues, concluded the fact that “a quick intervention can increase the rate of this germ elimination” [55,56]. The treatment schedule proposed by them consists of: rifampicin and TMP-SMX per os and mupirocin as topical treatment.

In patients with exacerbations, it is recommended to administer as a first line therapy: vancomycin and linezolid [56,57].

A cochrane meta-analyze publicized in 2018 by David Kh Lo *et co.*, about the early eradication of *MRSA* infection in cystic fibrosis patients, had demonstrated the superiority of eradication therapy through the presence of negative culture for *MRSA* at 28 days of treatment. Still, at 6 months from the end of treatment there were no significant differences observed between the treated group and the one on placebo [58].

The discussion about the long-term therapy in chronic infection with methicillin-sensitive or methicillin-resistant *Staphylococcus aureus* is still open. We cannot overlook two things when we decide to treat these infections: the possibility that our patient is only colonized with this commensal germ and the risk of a prophylactic therapy that can increase the germ resistance to antibiotics. Although the short-term benefits are well-known, the long-term effects still need to be studied.

All the antibiotics regimens are presented briefly in Table 3.

#### 3.1.4. *Burkholderia cepacia* Complex Strain

Although *Burkholderia complex* includes several strains, *Burkholderia cenocepacia*, *Burkholderia multivorans,* and *Burkholderia dolosa* are the most widespread in CF patients [11,59].

*Burkholderia* infection is associated with increased morbidity and marked reduction in life expectancy. There is an important heterogeneity in outcome among CF patients infected with *Burkholderia cepacia* complex: some patients have a significant deterioration of the lung function, with rapid clinical deterioration and decease [60], while others have *Burkholderia cepacia* complex for prolonged periods of time without any clinical manifestations. This noticeable variance in prognosis among CF patients has not been effectively elucidated but is suspected to be secondary to *Burkholderia cepacia* complex strains configurations and response to antibiotics [61].

The treatment of infections with *Burkholderia cepacia* complex strains is extremely challenging. *Burkholderia cenocepacia* is naturally resistant to most of β-lactams, polymyxins, aminoglycosides and can develop in vivo resistance to all antimicrobial classes [59].

In general, the use of trimethoprim–sulfamethoxazole is recommended. Besides newly instituted drugs such as aztreonam, doripenem, or high-dose tobramycin [62], the associations between first- and second-line agents are recommended [59]. Consequently, ceftazidime, meropenem, and penicillin (mainly piperacillin) are considered according to the in vitro antimicrobial susceptibility patterns [63]. Concerning penicillin (piperacillin–tazobactam and ticarcillin–clavulanate) in vitro susceptibility evaluation is mandatory before administration [62].

All the antibiotics regimens are presented briefly in Table 4.

#### 3.1.5. *Stenotrophomonas maltophilia*

*Stenotrophomonas maltophilia* is an aerobic Gram-negative germ, an emergent multi-drug resistant pathogen in CF patients, which forms a protective biofilm against antibiotics action, therefore making its treatment and eradication difficult. Being a resistant germ, chronic infection is possible, frequently associated with an almost three-fold increased risk of death or lung transplant among CF patients and an increase multidrug resistance [64].

It is uncertain whether *Stenotrophomonas maltophilia* infection is just a marker of severe lung disease or if the infection itself would lead to an acceleration of the illness development [65].

*Stenotrophomonas maltophilia* prevalence still varies considerably between CF centers [66], and the clinical implication in infections is variable [11]. Colonization with *Stenotrophomonas maltophilia* associates the occurrence of an immune response against the microorganism, with more reported exacerbations, but not additional advance in decline of respiratory function [67,68].

These species are resistant to numerous antibiotics and simply develop resistance to antibiotics during treatment. Susceptibility testing must therefore guide the choice of antibiotics, and combination therapy is usually recommended. Ceftazidime or carbapenems plus aminoglycosides are used, or aztreonam/ticarcillin/clavulanic acid combination therapy because of synergism against *Stenotrophomonas maltophilia*. Tetracyclines can be recommended, also ciprofloxacin with satisfactory but variable results and occurrence of resistance is frequent [11].

The efficiency of antibiotic treatments for this multi-drug resistant organism is still undecided, and despite several Cochrane reviews, there are no treatment recommendations [69], therefore further urgent recommendations are needed, considering *Stenotrophomonas maltophilia* aggressivity [65].

#### 3.1.6. Infection with *Achromobacter* (Alcaligenes) *xylosoxidans*

*Achromobacter xylosoxidans*, formerly known as *Alcaligenes denitrificans subspecia xylosoxidans*, is a type of Gram-negative bacillus, aerobic, mobile, non-fermenting glucose, which can be identified in soil and water [38]. *Achromobacter xylosoxidans* is intrinsically resistant to most antibiotics and often acquires in vitro resistance to additional antibiotics after exposure to antibiotics. The morphology of Achromobacter colonies shows similarities with the appearance of *Pseudomonas aeruginosa* colonies [70].

In the first infection, the therapy with colistin or cotrimoxazole intravenously is recommended, then continuing the therapy with colomycin inhaler for 3 months, or cotrimoxazole orally 1 month + colomycin inhaler 3 months. Exacerbation of chronically infected patients involves the use of two antipseudomonal antibiotics, and in case of chronic infection it is recommended for long-term inhaled colomycin [11].

A study published by Saiman et al., showed that certain strains of *Achromobacter xylosoxidans* had increased antibiotic resistance. Antibiotics such as minocycline, imipenem, meropenem, piperacillin, and piperacillin tazobactam had the highest activity and inhibited 55% of the strains. However, most of the tested strains demonstrated resistance to the remaining testing agents. The inhibition by higher concentrations of tobramycin and colistin was also studied, the results showing that most strains (97%) were resistant to conventional concentrations of tobramycin, but 41% of strains were inhibited by inhaled tobramycin concentrations. In this study, 8% of the studied *Achromobacter xylosoxidans* strains were resistant to high concentrations of colistin [71].

All the antibiotics regimens are presented briefly in Table 5.

#### 3.1.7. *Nontuberculous mycobacteria* (NTM)

Nontuberculous mycobacteria are commonly found in the environment. Prevalence of nontuberculous mycobacteria (NTM) infections is growing among CF population [72]. Nontuberculous mycobacteria species (most commonly *Mycobacterium avium* complex and *Mycobacterium abscessus*) are isolated from the respiratory tract of approximately 5% to 40% of individuals with cystic fibrosis; also, they cause a rapid lung function decline [73]. *Mycobacterium abscessus* especially is associated with worse outcome and the need for NTM treatment post transplantation and as a result, many transplant centers now consider the presence of NTM lung disease a contraindication for lung transplant [74].

MNTBs include several species, of which *Mycobacterium abscessus* complex has significant lung effects. NTM can be divided into fast-growing mycobacteria and a slower-growing group. The fast-growing mycobacteria is a part of the *Mycobacterium abscessus* complex and at the same time they seem to be more virulent than mycobacteria which show a slower growth [75]. The diagnosis involves the presence of at least two positive samples, taken from bronchoalveolar lavage or induced sputum culture. Rifampicin, ethambutol, and azithromycin may be effective for the *Mycobacterium avium* complex [76]. *Mycobacterium abscessus* infection benefits from intravenous antibiotic therapy with imipenem, amikacin, quinolone for 3 weeks and later, and an oral consolidation therapy: rifampicin, azithromycin, ethambutol +/- inhalation, duration: 12–24 months [77]. Sterilization is defined by four negative cultures during a year, after the end of treatment, eradication failure involves chronic administration of antibiotics in double combination [76].

The US and European CF NTM guidelines (2016) recommend in the initiation phase: azithromycin (oral) and intravenous amikacin, together with the combination of imipenem, tigecycline, cefoxitin, for 3–12 weeks. For the continuation phase the treatment includes the following drugs: azithromycin and inhaled amikacin with combination of two to three antibiotics: clofazimine, minocycline, moxifloxacin, or linezolid, for a period of time of 12–18 months [77,78].

In the study conducted by Alison DaCosta et al., for the CF patients treated for *Mycobacterium abscessus* respiratory infection, the following drugs were used: the most commonly used in the acute phase was amikacin, being administered intravenously in 68% patients and in 11% of cases was administrated by inhalation. Among beta-lactams, cefoxitin was the most commonly used in 76%. Concerning the class of macrolides, clarithromycin was used in 57% of cases, and azithromycin in 35%. Among the newer and less commonly used antibiotics, tigecycline was used in 32% and oral linezolid in 16%. More than half of the patients (59.4%) received three or more drugs and a “traditional” regimen that includes a beta-lactam antibiotic, a macrolide, and amikacin. Patients treated with clarithromycin had a higher bacterial clearance than those treated with azithromycin [75].

All the antibiotics regimens are presented briefly in Table 6.

#### 3.1.8. Infection with Haemophilus Influenza

*Haemophilus influenzae* and *Haemophilus parainfluenzae* are the most common species of *Haemophilus* that colonize the respiratory tract of children from an early age. Both species are frequently identified in the respiratory tract of children with CF, especially during exacerbation episodes [79]. Although many of the classic pathogens involved in lung disease with CF have been well studied, little is known about the role of Haemophilus species in the critical period of early childhood. A better understanding of *Haemophilus influenzae* and *Haemophilus parainfluenzae* infection is important because it is known that lung disease begins in early childhood and follows a linear progression.

For those who are diagnosed with *Hemophilus influenzae* infection for the first time, amoxicillin + clavulanic acid is recommended for 4 weeks. It can be combined with azithromycin 10 mg/kg body weight/day or clarithromycin 15–20 mg/kg body weight/day, depending on the patient’s clinical condition, for one month. Cefixime is recommended in the first treatment with *Hemophilus influenzae* resistant to first-line therapy, and in exacerbation: ceftazidime + amikacin or tobramycin, or amoxicillin + clavulanic acid (if not previously received), 14 days. Chronic infections are rare, being defined by the presence of two positive cultures/year and may benefit from the treatment with macrolide drugs such as azithromycin, which may also be recommended for its anti-inflammatory effect.

A recently published study specified the ability of *Hemophilus influenzae* to form a biofilm [80], thus presenting a high risk of developing resistance to the antibiotic therapy used by the first intention [81]. The study published by Watts et al. detailed the accumulation of resistance of *Hemophilus influenzae* and *Haemophilus parainfluenzae* to ampicillin (23.9%to 58.5% in *Hemophilus influenzae*, 13.2 to 50.0% *Haemophilus parainfluenzae*), and to cotrimoxazole (21.4% to 71.1% in *Hemophilus influenzae*, 14.9% to 44.2% in *Haemophilus parainfluenzae*) [82].

Resistance to cotrimoxazole (combination of trimethoprim and sulfamethoxazole) was similar in both species (35% in *Hemophilus influenzae*, 31% in *Haemophilus parainfluenzae*), but resistance to amoxicillin + clavulanic acid was found to be higher in *Haemophilus parainfluenzae* than in *Hemophilus influenzae*. Rifampicin resistance was found to be low in both species (17% in *Hemophilus influenzae*, 7% in *Haemophilus parainfluenzae*). Resistance to several drugs was also more commonly identified in *Haemophilus parainfluenzae* [82].

All the antibiotics regimens are presented briefly in Table 7.

## 4. Discussion

The microbiological spectrum of the infections found in cystic fibrosis patients is impressionably vast and, above all, consist of extremely resistant bacterial strains. Subsequently, the treatment of CF lung disease is not an easy task to accomplish, especially in children.

In conclusion, comparative efficacy research should be performed for the assessment of the most beneficial, individually applied, antibiotic therapy regimen. This would be challenging for cystic fibrosis population, considering the difficulty of choosing the correct antibiotic therapy in CF pulmonary infections. Therefore, an online therapy platform and coordinated treatment protocols resulting from clinical experience should be available for clinicians; also, further research should be continued regarding the most effective types of antibiotics and routes of administration, which could be remarkably beneficial to CF patients.

This is the reason why the personalization of antibiotics therapy among CF patients raises significant issues regarding the proper antibiotics’ regimen in certain microbial exposure, the task being an asymmetric challenge, with the balance more prone to the infection part (Appendix A).

## Figures and Tables

**Figure 1 children-09-01258-f001:**
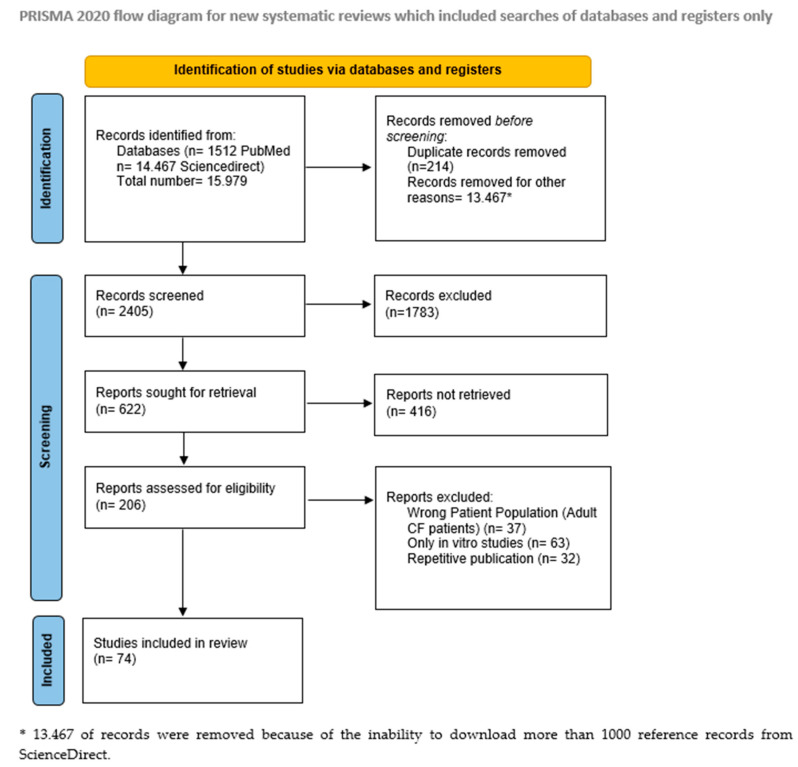
PRISMA flowchart of the literature screening.

**Table 1 children-09-01258-t001:** Recommended antibiotics for *Pseudomonas aeruginosa* infection in CF patients.

*Microbe*	Drug	Dose	Administration	Period	Reference
* Pseudomonas* *aeruginosa* First infection Exacerbation	Ciprofloxacin + Tobramycin Or Colomycin Ceftazidime or Meropenem or Colomycin + Amikacin Or Tobramycin	15–20 mg/kgc/day 300 mg/kgc/day 75–150 mg/kgc/day 150–200 mg/kg/day in 3 or 4 divided doses 120 mg/kg/day 25.000 Units/kg 30 mg/day 10 mg/kg/day	*oral* *nebulized* *nebulized* *i.v.*	14 days 28 days/month, 28 days break, 3 months daily, 6 months 14–21 days	Döring et al. [14] Flume et al. [9] Treggiari et al. [44]

**Table 2 children-09-01258-t002:** Recommended antibiotics for *Staphylococcus aureus* infection in CF patients.

*Microbe*	Drug	Dose	Administration	Period	Administrations per Day	Reference
*Staphylococcus* *aureus* First infection	Flucloxacillin Fusidic acid Rifampicin	100 mg/kg/day 25–50 mg/kg/day 15–20 mg/kg/day	*oral* *oral, i.v.* *oral, i.v.*	2–4 weeks	3–4 2–3 2	Döring et al. [11]

**Table 3 children-09-01258-t003:** Recommended antibiotics for *Staphylococcus aureus MRSA* infection in CF patients.

*Microbe*	Drug	Dose	Administration	Period	Administrations per Day	Reference
*Staphylococcus* *aureus* *MRSA* First infection	Fusidic acid	25–50 mg/kg/day	*oral*	2–4 weeks	2–3	Döring et al. [11] Dolce et al. [55] Waterer et al. [52] Chmiel et al. [5]
	Rifampicin	15–20 mg/kg/day	*oral*	2–4 weeks	2–3	
	TMP-SMX	8–12 mg TMP/kg/day	*oral*	2–4 weeks	2	
	Vancomycin +	250 mg nebulized	*nebulized*	28 days/month	2	
	Mupirocin		*topical (intranasal)*	14 days	2	
Exacerbation	Vancomycin or Linezolid	15–20 mg/kg Q6–8 <12 years: 10 mg/kg Q8 >12 years: 10 mg/kg Q12	*i.v.* *i.v.*	14–21	3–4 2–3 (<45 kg consider Q8 hour dosing)	

**Table 4 children-09-01258-t004:** Recommended antibiotics for *Burkholderia cepacia complex* infection in CF patients.

Microbe	Drug	Dose (mg/kg/Day)	Administration	Period	Administrations per Day	Reference
*Burkholderia* *cepacia complex*	TMP-SMX Aztreonam Tobramycin Piperacillin/tazobactam Ceftazidime Ticarcillin–clavulanate	50–100 *(oral)* 10–20 *(i.v.)* 150–250 10 350–450 50–200 200–300/6–10	*oral* or *i.v.* *i.v.* *i.v.* *i.v.* *i.v.* *i.v.*	2–4 weeks	2–4 3 1–3 4 3 3	Döring et al. [11]

**Table 5 children-09-01258-t005:** Recommended antibiotics for *Stenotrophomonas maltophilia* and *Achromobacter xylosoxidans* infection in CF patients.

Microbe	Drug	Dose mg/kg/Day	Administration	Period	Reference
*Stenotrophomonas maltophilia and* *Achromobacter* *xylosoxidans*	Minocycline	2–3	*oral*	2–4 weeks	Döring et al. [11]
	Ceftazidime	150–200	*i.v.*		
	Meropenem	120	*i.v.*		
	Ciprofloxacin	20–30	*oral, i.v.*		
	Aztreonam	150–250	*i.v.*		
	Amikacin	30	*i.v.*		
	Doxycycline	2–3	*oral*		
	TMP-SMX	50–100 *(oral)* 10–20 *(i.v.)*	*oral, i.v.*		
	Ceftazidime	150–200	*i.v.*		
	Meropenem	120	*i.v.*		
	Colomycin	25.000 Units/kg	*i.v.*		
	Tobramycin	10	*i.v.*		
	Ciprofloxacin	20–30	*oral, i.v.*		
	Aztreonam	150–250	*i.v.*		
	Piperacillin/tazobactam	350–450	*i.v.*		

**Table 6 children-09-01258-t006:** Recommended antibiotics for *Nontuberculous mycobacteria* infection in CF patients.

*Microbe*	Drug	Dose	Administration	Period	Reference
*Mycobacterium avium complex*	Clarithromycin Azithromycin Ethambutol Rifabutin Rifampicin Amikacin	15 mg/kg 5 mg/kg 15 mg/kg 150–300 mg/day 450–600 mg/day 15 mg/kg/day	*oral* *oral* *oral* *oral* *oral* *i.v.*	2–4 weeks	Döring et al. [11] Chmiel et al. [74] Floto et al. [77]
*Mycobacterium abscessus*	Imipenem + Amikacin after 3 weeks oral consolidation with: Rifampicin Azithromycin Ethambutol	20–25mg/kg Q6 15 mg/kg/day	*i.v.* *i.v.* *oral*	21 days 14 days 12–24 months	

**Table 7 children-09-01258-t007:** Recommended antibiotics for *Haemophilus influenzae* infection in CF patients.

*Microbe*	Drug	Dose mg/kg/Day	Administration	Period	Reference
*Haemophilus influenzae*					Döring et al. [11]
First infection	Amoxicillin + clavulanic acid	50–100	*oral*	4 weeks	
	Azithromycin	10	*oral*	4 weeks	
	Clarithromycin	15–20	*oral*	4weeks	
	Cefixime	8–16	*oral*	2–4 weeks	
2. Exacerbation	Ceftazidime + Amikacin or Tobramycin	150–200 15 10	*i.v.*	14 days	

## Data Availability

Not applicable.

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
