# Peer review of "Antibiotherapy in Children with Cystic Fibrosis—An Extensive Review"

_children, 2022, doi:10.3390/children9081258_

Round 1

Reviewer 1 Report

Thank you for the opportunity to review this manuscript. It is a very important topic and the authors have done an admirable job of covering a large topic. I think the manuscript needs major revision prior to being ready for publication:

1. Revision of language to improve the writing (I have attached a file with edits to the abstract, but similar changes need to be made for entire document)

2. The manuscript currently sits somewhere between an narrative review and a systematic review. If wanting to continue to pursue publication as a systematic review then more description of methods used post database search is needed. In the result I would not describe the entire background re each organism, but focus only on trials comparing antibiotic treatment regimes in more detail 

Reviewer 2 Report

Review Antibiotherapy in cystic fibrosis: an extensive review

The work is interesting; in turn, the issue is very important given its high impact on human health. The methodology used is good, only that a summary of the results could be made with a table or in other ways; a good integration in each case.

On the other hand, better working the results will allow a more engaged discussion.

Additional comments:

Cystic fibrosis is a disease of high frequency and marked polymorphism, which mainly affects the lung and pancreas. This disease of difficult prognosis has among its consequences the risks of bacterial infections for which specific and prolonged antibiotic treatments are required. Antibiotherapy is one of the main treatments in cystic fibrosis. These would be the reasons why this review addresses antibiotherapy by doing a review of it.
The topic is relevant and the interesting thing is to be able to carry out a complete review, since in general the works deal with the treatment when the patient is affected by a specific type of germs.

The methodology used can be better described or detailed, and the inclusion and exclusion criteria should be placed separately.

It is convenient to present tables where each antibiotic is indicated and its properties or advantages and disadvantages of use and any special features.

By adjusting the results, or improving the presentation, it will facilitate the presentation of the conclusions. References are appropriate.

Reviewer 3 Report

The article written by Ciuca et al. aims to review the existing literature on antibiotic therapy in cystic fibrosis. The topic is high relevant today and of great interest but the article is weak from many points of view. First of all, the methodological rigor: the criteria of the systematic review (PRISMA guidelines) are not fully respected; we are facing a narrative review. It is not clear if we are referring to antibiotics in pediatric age (which would suggest an article “children” journal) or in general. Even the treatment of the individual paragraphs is not very detailed and not updated but above all not very useful from a practical point of view as nothing new emerges compared to what is already known. Specifically, some ideas for improvement:

• The first main point concerns the subject of the review. It is not clear if you want to deepen the aspect relating to children (which the journal would assume) but there is no trace of this either in the introduction or in the title. It is only indicated in the section relating to the methods that studies involving "a number of pediatric CF" have been included. In this regard, it should be specified what is meant by "a number" and by "pediatric" (0-14? 0-16? 0-18?).

• The section on methods is absolutely lacking. The authors report that the PRISMA guidelines were followed. However, it doesn't seem totally correct. To this end, it would be advisable to include the PRISMA checklist also as supplementary material or to avoid talking about systematic reviews.

• The inclusion and exclusion criteria should be separated. Above all, it is not clear on the basis of which elements a study was considered valid or not. It seems a single author's judgment not validated by predetermined criteria; this is not in line with the PRISMA guidelines.

• The name of the bacteria should be standardized. In the text we find: Pseudomonas aeruginosa, Pseudomonas aeruginosa, P. aeruginosa

• Regarding the treatment of individual microorganisms, no mention is made of the specific applications of antibiotics in children. For example, in the treatment of Pseudomonas aeruginosa infection it must be specified that the use of inhaled aminoglycosides is not always allowed / recommended in younger children.

• Staphylococcus aureus infection: I believe that paragraph 3.2 deals with methicillin-sensitive staphylococcus (MSSA) and not MRSA as indicated in line 229. In the case of MSSA, the main point is whether to treat it or not. From the discussion of the paragraph, it is not clear if and how to treat this pathogen.

• Stenotrophomonas maltophilia: no reference is made to the use of trimethoprim – sulfamethoxazole which is the antibiotic that is usually used in these cases. Please clarify.

• All antibiotics: no section refers to the specific use of certain antibiotics in children. Pros cons. In the methods it is indicated that reference would have been made mainly to pediatric studies but there is no trace of all this.

• The discussion is meager. A clear message does not emerge, there is no talk of future prospects.

• No section refers to the new combinations of antibiotics (ceftazidime / avibactam; ceftolozane / tazobactam, ceftarolin ...)

• There is no table summarizing the main antibiotic treatments against the main microorganisms.

Round 2

Reviewer 3 Report

The authors responded comprehensively to the numerous objections of the first revision and significantly improved the manuscript which, now, can be considered for publication.

I suggest reviewing the manuscript for typos.

Author Response

We have correct our typos. Thank you for your observation.